# Mechanical Fracture and Fatigue Characteristics of Fine-Grained Composite Based on Sodium Hydroxide-Activated Slag Cured under High Relative Humidity

Hana Šimonová [1,*], Barbara Kucharczyková [1], Vlastimil Bílek [2], Lucie Malíková [1], Petr Miarka [1] and Martin Lipowczan [1]

1 Faculty of Civil Engineering, Brno University of Technology, Veveří 331/95, 602 00 Brno, Czech Republic; barbara.kucharczykova@vutbr.cz (B.K.); malikova.l@fce.vutbr.cz (L.M.); petr.miarka@vut.cz (P.M.); lipowczan.m@fce.vutbr.cz (M.L.)

2 Faculty of Chemistry, Brno University of Technology, Purkyňova 464/118, 612 00 Brno, Czech Republic; bilek@fch.vut.cz

* Correspondence: simonova.h@vutbr.cz; Tel.: +420-541-147-381

**Abstract:** A typical example of an alternative binder to commonly used Portland cement is alkali-activated binders that have high potential as a part of a toolkit for sustainable construction materials. One group of these materials is alkali-activated slag. There is a lack of information about its long-term properties. In addition, its mechanical properties are characterized most often in terms of compressive strength; however, it is not sensitive enough to sufficiently cover the changes in microstructure such as microcracking, and thus, it poses a potential risk for practical utilization. Consequently, the present study deals with the determination of long-term mechanical fracture and fatigue parameters of the fine-grained composites based on this interesting binder. The mechanical fracture parameters are primarily obtained through the direct evaluation of fracture test data via the effective crack model, the work-of-fracture method, the double-*K* fracture model, and complemented by parameter identification using the inverse analysis. The outcome of cyclic/fatigue fracture tests is represented by a Wöhler curve. The results presented in this article represent the complex information about material behavior and valuable input parameters for material models used for numerical simulations of crack propagation in this quasi-brittle material.

**Keywords:** fracture; fatigue; experiment; slag; alkali activation; fine-grained composite

## 1. Introduction

Although Portland cement and its blends are and in the near future will be the most common binders in practical applications around the world because of their availability, versatility, and tradition, there are increasing efforts to search for some alternative binders. A typical example is alkali-activated binders that have high potential as a part of a toolkit for sustainable construction materials, especially in applications where resistance to elevated temperatures or aggressive environments is required [1]. One group of alkali-activated materials is alkali-activated slag (AAS), which sets and hardens rapidly even at room temperatures and reaches high strengths [2]. On the other hand, AAS suffers from extensive shrinkage accompanied by cracking and was found to be very brittle [3]. There is a lack of information about the long-term properties, fracture characteristics, or fatigue of AAS. In addition, its behavior is characterized most often in terms of its compressive strength; however, this is not sensitive enough to sufficiently cover the changes in microstructure such as microcracking, and thus, it poses a potential risk for practical utilization. Consequently, the present study deals with the determination of long-term mechanical fracture and fatigue parameters of the fine-grained composites based on this interesting binder. To record and evaluate the formation of cracks and their further propagation during the

mechanical fracture tests, it is necessary to design an AAS material with acceptable material characteristics (sufficient compressive and tensile strength, and appropriate value of modulus of elasticity) together with a low potential of cracking. This is rather a challenge in the case of AAS materials for in situ applications with curing under ambient conditions. It was proven that their composition is sensitive to curing conditions, and different behavior can be expected for the different type and dosage of activator. Our own experiment is presented as an example. Two AAS materials of a similar composition containing different activators were produced and cured under wet conditions with relative humidity (RH) $\geq$ 95% to observe cracking tendency. Despite the high relative humidity during curing, opened cracks appeared in the case of AAS material activated by waterglass (see Figure 1).

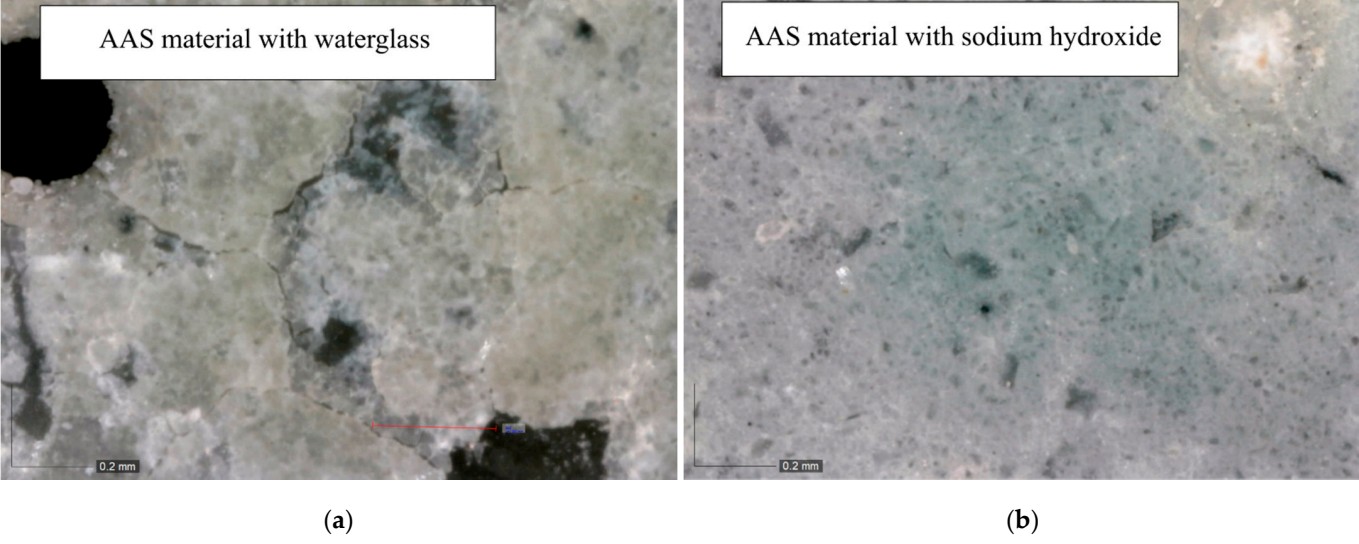

(**a**)　　　　　　　　　　　　　　　　　　　　　　　　　(**b**)

**Figure 1.** The surface of the alkali-activated slag (AAS) materials with a different activator (waterglass (**a**) and sodium hydroxide (**b**)) cured in relative humidity (RH) $\geq$ 95% (240× magnification).

Assessment of the fracture behavior of quasi-brittle materials is performed for notched specimens/structures subjected to mode I of loading. The investigations of the fracture phenomena such as the relation of the fracture process zone (FPZ) extent to the size/geometry/boundary effect, etc., in other modes of loading and/or their combination (mixed-modes of loading) are studied rather rarely despite the fact that the real structures made of quasi-brittle materials in civil engineering applications are usually loaded in a mixed-mode manner. It should also be emphasized that many structures are often subjected not only to static but also to repetitive cyclic loads of high-stress amplitude. Examples of such cyclic loads include automotive and train traffic, machine vibration, and wind action. The processes occurring within the quasi-brittle material structure and leading to its degradation under cyclic loading are more complicated in comparison to those affecting metals [4]. That is the reason why knowledge of the behavior of quasi-brittle materials, not only under static or quasi-static conditions but also under cyclic conditions, is very important for the complex description of crack propagation in such materials. However, fatigue tests of quasi-brittle materials and structures are expensive, and for this reason, numerical modeling [5,6] can represent an effective tool for the prediction of the damage process and fatigue life of such materials under different service conditions. For the effective and correct use of a numerical (material) model, it is often necessary to tune its parameters using data obtained from experimental measurements. The correct evaluation of such data is becoming a prerequisite for the correct use of numerical models in practice.

Therefore, this research area was covered by the currently solved research project "Advanced characterization of cracks propagation in composites based on alkali-activated matrix". An extensive numerical and experimental analysis of the fracture behavior of

selected composites with an alkali-activated matrix subjected to static (also in the mixed-mode) and cyclic/fatigue loading was performed. For a description of the crack propagation under mixed-mode I/II conditions, various test configurations were designed based on the literature search of test specimens used for testing the fracture parameters of rocks and cement-based composites. For this purpose, semi-circular disc specimens in three-point bending [7–11], centrally cracked Brazilian discs [10,12,13], and prismatic-specimens in asymmetric three- [14,15] or four-point [14,16] bending configurations are mostly used. Semi-circular discs were chosen for the present study. The type of specimens was chosen with respect to the practical application to the in situ structures when drilled cores are usually taken for the verification of real mechanical characteristics. Selected semi-circular specimens with an inclined initial notch were loaded in a three-point bending configuration (SCB) for the purpose of experimental verification of numerical simulations of crack propagation under mixed-mode I/II condition; see the selected configuration in Figure 2a. Note that numerical simulation was performed prior to the experimental verification to suggest different geometrical configurations with various mixed-mode I/II conditions (to cover as wide a range of mixed-mode levels as possible) and to design the experiment. The detailed information can be found in a separate article, which is currently under consideration for publication [17].

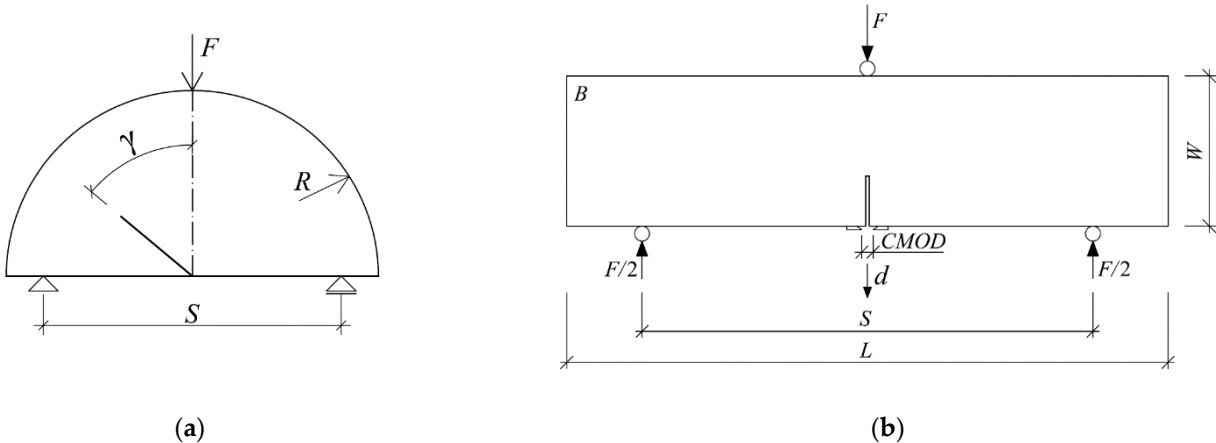

(**a**)  (**b**)

**Figure 2.** Three-point bending fracture test configuration of a selected semi-circular disc specimen with an inclined initial notch (**a**); and a beam specimen with a central edge notch (**b**).

The numerical modeling of the fracture initiation and propagation in quasi-brittle materials has been developed for several decades. Impressive results have been achieved with continuous models (usually with the help of the finite element method (FEM)) enhanced by non-linear constitutive relations where, after reaching the tensile strength, the stress decreases gradually with the crack opening. If the model does not contain information about material meso-structure directly, spurious sensitivity of the results to mesh discretization appears [18]. Such mesh dependency must be eliminated with help of a localization limiter (crack band model, non-local model, gradient model). The continuous models are typically studied on simple experiments such as a three-point bending test, but they achieved excellent results also in cases of complex mixed-mode configurations [19]. When some of the models described above shall be applied, several important mechanical fracture parameters need to be determined/known. One of the fundamental fracture parameters is the fracture energy representing cracking resistance and fracture toughness. To be able to estimate the FPZ size and shape, the necessary parameters (such as tensile strength, fracture energy, stress−strain diagram, etc.), methods, and approaches have to be known.

Commonly used standard test specimens (prisms with nominal dimensions of $40 \times 40 \times 160$ mm$^3$) were manufactured and tested under typical configurations to obtain the basic physical, mechanical, fracture, and fatigue characteristics essential for numeri-

cal simulations. The test specimens were provided with an initial notch before the test, and subsequently, static and cyclic/fatigue fracture tests were performed in a three-point bending configuration; see Figure 2b.

The outcome of each static fracture test is a vertical force vs. midspan deflection and vertical force vs. crack mouth opening displacement diagram. The mechanical fracture parameters are primarily obtained through the direct evaluation of fracture test data via the effective crack model [20], the work-of-fracture method [21], and the double-*K* fracture model [22]. The vertical force vs. midspan deflection diagrams are also used to obtain mechanical fracture parameters indirectly—based on a combination of fracture testing and inverse analysis [23]. The inverse analysis has several advantages, such as its ability to identify parameters whose direct testing and evaluation are difficult or even impossible. The representative of such parameters is the tensile strength. The outcome of cyclic/fatigue fracture tests is represented by a Wöhler curve [4].

For the present study, sodium hydroxide was used as an alkaline activator to produce appropriate test specimens with low potential of cracking. It was experimentally verified by authors (see Figure 1) and confirmed by other researchers that such AAS material reaches sufficiently high strengths together with noticeably lower shrinkage [24,25], and thus, the risk of cracking before the test is reduced. The complex information about the material included the mechanical, fracture, and fatigue characteristics of AAS fine-grained composite used as the input values for a numerical simulation are presented herein. The standard test specimens with nominal dimensions of $40 \times 40 \times 160$ mm$^3$ were manufactured and tested under typical configurations. Non-traditional curing conditions of ambient temperature and high relative humidity (RH) $\geq$ 95% for a long period was chosen to prevent the cracking of the material before fracture and fatigue tests. In practical applications of investigated material, these non-traditional curing conditions can be found in the design of precast elements rather than in in situ applications.

## 2. Materials and Methods

### 2.1. Materials and Specimens

A fine-grained composite that used sodium hydroxide-activated slag as a binder and CEN siliceous sand with a maximum grain size of 2 mm (meeting requirements of EN 196-1) in the dose three times higher than the slag weight was produced. Granulated blast furnace slag with a predominant amount of glassy phase ground to the Blaine fineness of $400$ m$^2$/kg was used. It was alkali-activated using sodium hydroxide solution in the dose corresponding to 6% Na$_2$O of the slag weight. The water-to-slag ratio including water from an activator as well as extra-added water was 0.45. In addition, 1% of lignosulfonate-based plasticizer was used.

The mixing was carried out using a Hobart mixer. First, slag was mixed with liquid components, and then, the sand was gradually added. The total mixing time was 10 min. Then, the fresh mortar was cast into the molds and sealed with stretch polyethylene (PE) foil to prevent moisture loss (Figure 3a). After 24 h, the specimens were demolded and stored under laboratory conditions with the temperature of $21 \pm 2$ °C in a closed box with RH > 95% (Figure 3b) up to the date the particular tests were performed. There was an exception in the curing conditions for the specimens intended for shrinkage measurement. To observe the cracking tendency under dry-air curing conditions, the specimens were not protected from drying during the whole time of aging (including the first 24 h after molding) and stored in an air-conditioned laboratory with a temperature of $21 \pm 2$ °C and RH of $55 \pm 10$%. In this experiment, it is considered as the most critical curing with the risk of cracking.

In total, 72 prismatic specimens with nominal dimensions of $40 \times 40 \times 160$ mm$^3$ were manufactured for the experiment. The test specimens were manufactured in three batches. The fracture tests of prismatic specimens were performed at the age of 40, 160, and 530 days. The cyclic/fatigue fracture tests were performed between the age of 140 to 160 days. Shrinkage process and the development of dynamic modulus of elasticity were

measured in regular intervals to record the increase in these parameters during the whole duration of the experiment.

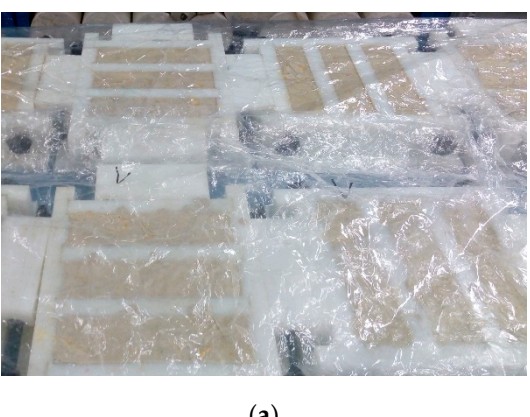

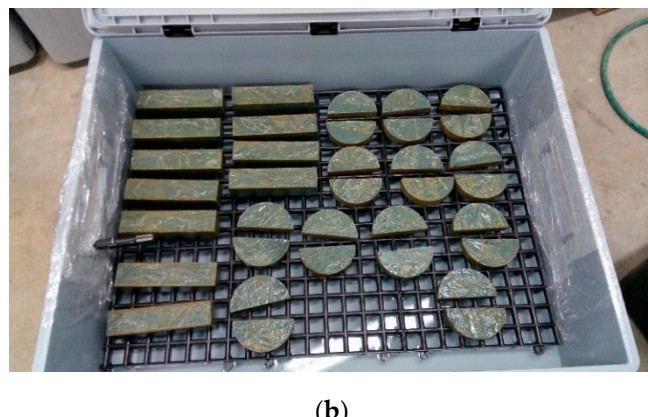

(**a**)                                   (**b**)

**Figure 3.** Manufactured test specimens covered with a stretch PE foil (**a**); test specimens after demolding prepared to storage in a closed box with RH > 95% (**b**).

### 2.2. Shrinkage Measurement

Prismatic specimens with dimensions of $40 \times 40 \times 160$ mm$^3$ equipped with stainless steel markers embedded into the ends of specimens were used for shrinkage measurement. The measurement set contained three test specimens. The measurement started immediately after demolding of the specimens at the age of 24 h and continued at regular intervals until the time when the length changes and mass losses were stabilized. The changes in length were measured using the measurement gauge equipped with the digital probe with a resolution of 0.001 mm. Simultaneously, the mass of each specimen was recorded using the scale with a resolution of 0.1 g. The specimens were stored in an air-conditioned laboratory with a temperature of $(21 \pm 2)$ °C and RH of $(55 \pm 10)$% during the whole time of measurement.

### 2.3. Dynamic Modulus of Elasticity and Rigidity, and Dynamic Poisson's Ratio

A non-destructive test based on the resonance method was employed to monitor the development of the dynamic modulus of elasticity $E_{rL}$ and dynamic Poisson's ratio $\mu_r$ of investigated material during aging. The specimens were stored at laboratory conditions with a temperature of $21 \pm 2$ °C in a closed box with RH > 95% (Figure 3b) during the whole time of measurement. The natural frequency of longitudinal and torsional vibrations was measured on the specimens with dimensions of $40 \times 40 \times 160$ mm$^3$ using a Handyscope HS4 oscilloscope equipped with an acoustic sensor. For more details about the principle of measurement, refer to [26]. The absolute values of $E_{rL}$ and $\mu_r$ were calculated in compliance with the ASTM C215-19 [27] as:

$$E_{rL} = 4 \left( \frac{L}{WB} \right) m f_L^2 \tag{1}$$

where $E_{rL}$ is the dynamic modulus of elasticity, $L$ is the length of specimens, $W$ and $B$ are cross-section dimensions (see Figure 2b), $m$ is the mass of specimens, and $f_L$ is the fundamental longitudinal frequency.

$$\mu_r = \left( \frac{E_{rL}}{2 \cdot G_r} \right) - 1 \tag{2}$$

where $\mu_r$ is the dimensionless dynamic Poisson's ratio, $E_{rL}$ is the dynamic modulus of elasticity, and $G_r$ is the dynamic modulus of rigidity, which was calculated as:

$$G_r = 4\left(\frac{LR}{WB}\right)mf_t^2 \tag{3}$$

where $R$ is the shape factor (1.183 for a square cross-section prism), and $f_t$ is the fundamental torsional frequency.

### 2.4. Static Fracture Tests

As mentioned above, three-point bending (3PB) tests were carried out to determine the selected mechanical fracture characteristics of the investigated composite with alkali-activated slag matrix. The above-mentioned prismatic specimens were provided with an initial notch one day before fracture tests were performed. The central edge notch with a nominal depth of about 1/3 of the specimen's height was made by a diamond blade saw. The span length was set to 120 mm. The specimens were tested at different ages of their hardening, namely at the age of 40, 160, and 530 days. Fracture tests were conducted using the stiff multi-purpose mechanical testing machine LabTest 6.250 with the load range of 0−250 kN. An arrangement of a static 3 PB fracture test is shown in Figure 4a. The loading process was governed by a constant increment of displacement of 0.02 mm/min during the whole course of testing. The vertical displacement (midspan deflection) $d$ was measured using the inductive sensor mounted in a special measurement frame placed on the specimens (see Figure 4b). Crack mouth opening displacement ($CMOD$) was measured using a strain gauge mounted between blades fixed on the bottom surface of the test specimens, close to the initial notch (see Figure 4b).

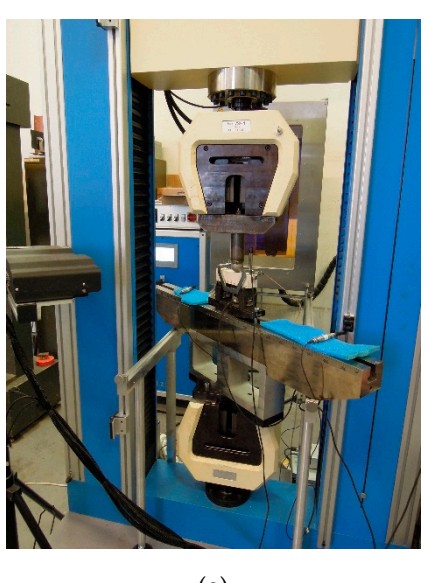

(**a**)

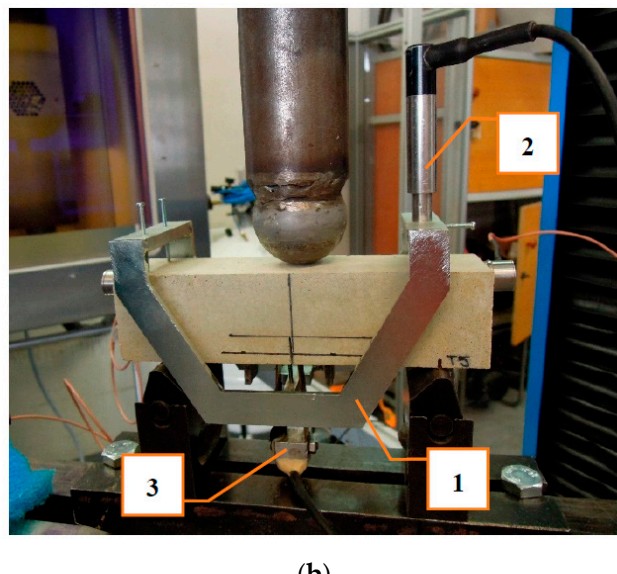

(**b**)

**Figure 4.** Arrangement of static fracture test (**a**), fracture test configuration in three-point bending (3PB) (**b**) including the measurement frame (1), and the sensors for measuring the midspan deflection (2) and the crack mouth opening displacement (3).

The outcome of each test is a vertical force $F$ vs. $d$ diagram and $F$ vs. $CMOD$ diagram. The diagrams recorded during the tests were processed to obtain the correct input values for consecutive diagram evaluations using the selected fracture models described below. In this case, the modification of diagrams covered the elimination of duplicate points, the shifting of the origin of the coordinate system, the smoothing of the diagram, and the reduction of the number of points. The processing of recorded diagrams was performed in GTDiPS software [28], which is based on advanced transformation methods used for the processing of extensive point sequences. The $F−d$ and $F−CMOD$ diagrams after the above-mentioned corrections for investigated ages of specimens are introduced in Figure 5.

Note that in the case of $F-CMOD$ diagrams, only their part until a peak load is used for consecutive evaluation. Therefore, only the details of these parts are introduced in Figure 5d–f.

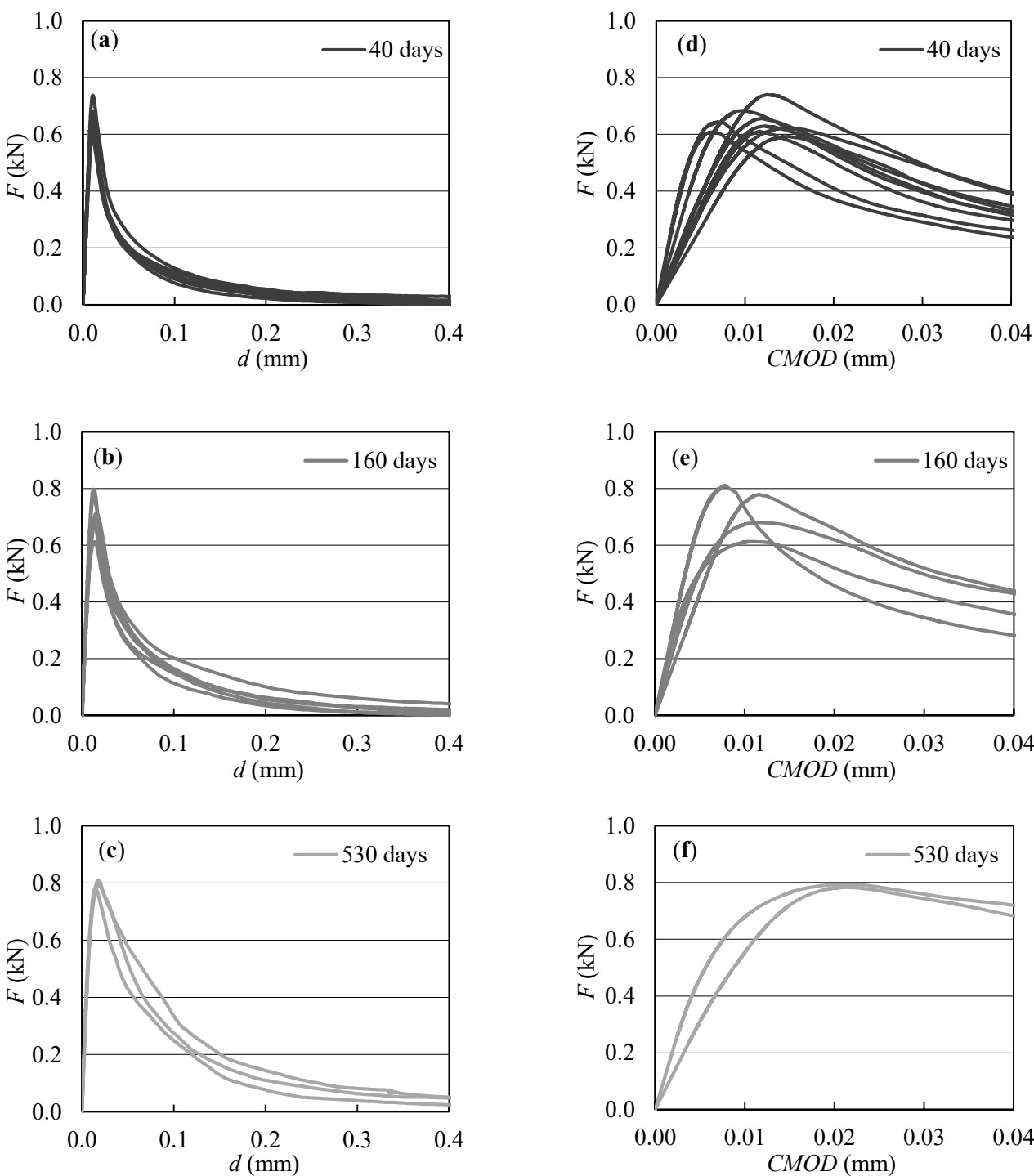

**Figure 5.** $F-d$ diagrams (**a**–**c**) and $F-CMOD$ diagrams (**d**–**f**) from 3PB fracture test for investigated ages of specimens: 40, 160 and 530 days, respectively.

2.4.1. Direct Evaluation of Mechanical Fracture Parameters from $F-d$ Diagrams

The value of static modulus of elasticity was calculated from the initial part of the recorded $F-d$ diagrams according to Stibor [29]:

$$E = \frac{F_i}{4Bd_i}\left(\frac{S}{W}\right)^3\left[1 - 0.387\frac{W}{S} + 12.13\left(\frac{W}{S}\right)^{2.5}\right] + \frac{9}{2}\frac{F_i}{Bd_i}\left(\frac{S}{W}\right)^2 F_1(\alpha_0) \tag{4}$$

where $F_i$ is the vertical force in the ascending linear part of the diagram, $B$ is the specimen width, $d_i$ is the midspan deflection corresponding to the force $F_i$, $W$ is the specimen height, $S$ is the span length (see Figure 2b), and

$$F_1(\alpha_0) = \int_0^{\alpha_0} xY^2(x)dx \tag{5}$$

where $\alpha_0 = a_0/W$ is the relative notch depth ($a_0$ is the initial notch depth), and $Y(x)$ is the geometry function for the 3PB configuration proposed by Brown and Strawley [20]:

$$Y(\alpha_0) = 1.93 - 3.07\alpha_0 + 14.53\alpha_0{}^2 - 25.11\alpha_0{}^3 + 25.80\alpha_0{}^4. \tag{6}$$

Several adaptations of linear elastic fracture mechanics (LEFM) have been proposed to consider the non-linear behavior of quasi-brittle materials in an approximate manner. One representative is an effective crack model (ECM) [20], which includes the effect of pre-peak nonlinear behavior of a real quasi-brittle structure containing initial notch through an equivalent elastic structure containing a notch of effective length $a_e > a_0$. The effective crack length $a_e$ is calculated from the secant stiffness of the specimen corresponding to the peak load $F_{\max}$ and matching midspan deflection $d_{F\max}$. The $a_e$ for the prismatic specimen with a central edge notch tested in the 3 PB configuration was determined according to [20]:

$$d_{F\max} = \frac{F_{\max}}{4BE}\left(\frac{S}{W}\right)^3\left[1 + \frac{5qS}{8F_{\max}} + \left(\frac{W}{S}\right)^2\left\{2.70 + 1.35\frac{qS}{F_{\max}}\right\} - 0.84\left(\frac{W}{S}\right)^3\right] + \frac{9}{2}\frac{F_{\max}}{BE}\left(1 + \frac{qS}{2F_{\max}}\right)\left(\frac{S}{W}\right)^2 F_1(\alpha_e), \tag{7}$$

where $q$ is the self-weight of the specimens per unit length.

Subsequently, the effective fracture toughness value $K_{Ice}$ was calculated using a LEFM formula according to Karihaloo [20]:

$$K_{Ice} = \frac{6M_{\max}S}{BW^2}Y(\alpha_e)\sqrt{a_e}, \tag{8}$$

where $Y(\alpha_e)$ is the geometry function (6) with $\alpha_e = a_e/W$ [20], and $M_{\max}$ is the bending moment due to the maximum applied vertical force $F_{\max}$ and self-weight of the specimen.

According to the RILEM recommendation [21], the specific fracture energy $G_f$ is the average energy given by dividing the total work of fracture by the projected fracture area (i.e., the area of the initially uncracked ligament). Therefore, for a specimen of depth $W$ and initial notch length $a_0$, the fracture energy $G_f$ is given by:

$$G_f = \frac{1}{(W - a_0)B}\left(\int Fdd + m_q d_{\max}\right), \tag{9}$$

where $m_q$ is the specimen weight, $d$ is the midspan deflection, and $d_{\max}$ is the maximum vertical deflection at failure.

Note that the informative compressive strength $f_c$ was determined according to ČSN EN 196-1 [30] on the fragments remaining after the fracture experiments had been conducted. The maximum vertical force $F_{\max}$ recorded during the fracture test was used for calculation of the informative flexural strength:

$$f_f = \frac{3F_{\max}S}{2B(W - a_0)^2}. \tag{10}$$

### 2.4.2. Identification of Mechanical Fracture Parameters from $F-d$ Diagrams

The artificial neural network-based inverse analysis method was employed to identify selected mechanical fracture parameters from $F-d$ diagrams recorded during 3PB static fracture tests. The inverse procedure proposed by Novák and Lehký [23] transforms fracture test response data into the desired mechanical fracture parameters. This procedure is based on a comparison of the experimentally recorded $F-d$ diagrams with the results obtained by simulating the 3PB test numerically. The artificial neural network (ANN) is used here as a substitute model of an unknown inverse function between input mechanical fracture parameters and corresponding response parameters.

The dataset intended for training the ANN was prepared numerically using a finite element method (FEM) model. The simulations represent a static 3PB fracture test (see Figure 2b) with random realizations of material parameters. The random values of material parameters were generated using the stratified sampling method and by performing an inverse transformation of the distribution function to reflect the probability distribution of the parameter. In this case, the ATENA FEM program (Červenka et al. [31]) was used for the numerical simulation of the 3PB fracture test. The 3D NonLinear Cementitious 2 material model was selected to govern the gradual evolution of localized damage.

The used identification system comprises of an ensemble of ANNs. Three-dimensional space is defined by three mechanical fracture parameters—modulus of elasticity $E_{\mathrm{ID}}$, tensile strength $f_{\mathrm{t,ID}}$, and specific fracture energy $G_{\mathrm{f,ID}}$. The whole space is divided into several subspaces because of the various materials properties of fine-grained composites with different binders. Each subspace covers a single robust ANN trained for a limited range of parameters. A suitable subspace for the particular analyzed specimen is selected automatically, and the relevant ANN is activated based on an initial analysis of the fracture test response data. The above-mentioned mechanical fracture parameters are calculated by simulating the ANN with obtained response parameters. For more details on ANN-based identification and utilization of the ANN ensemble, see [32,33].

### 2.4.3. Direct Evaluation of Mechanical Fracture Parameters from $F-CMOD$ Diagrams

The double-$K$ fracture (2K) model [22] was used for the evaluation of the $F-CMOD$ diagrams to determine selected fracture parameters. The parameters describing different phases of the fracture process were determined using this fracture model. The unstable fracture toughness $K_{\mathrm{Ic}}{}^{\mathrm{un}}$ is defined as the critical stress intensity factor corresponding to the maximum load, and it represents the phase of unstable crack propagation. This parameter is similar to effective fracture toughness (Equation (8)) used in the effective crack model by Karihaloo [20]. The equivalent elastic crack length $a_c$ was determined from the following equation [22]:

$$CMOD_{F_{\max}} = \frac{6F_{\max}Sa_c}{BW^2E}V(\alpha_c) \tag{11}$$

where $CMOD_{F\max}$ is $CMOD$ corresponding to peak load $F_{\max}$, and

$$V(\alpha_c) = 0.76 - 2.28\alpha_c + 3.87\alpha_c{}^2 - 2.04\alpha_c{}^3 \frac{0.66}{(1 - \alpha_c)^2} \tag{12}$$

where $\alpha_c = (a_c + H_0)/(W + H_0)$; $H_0$ is the thickness of blades fixed on the bottom surface of the specimens between which the strain gauge was placed.

When the equivalent elastic crack length $a_c$ is known, the $K_{\mathrm{Ic}}{}^{\mathrm{un}}$ was determined according to Equation (8), when the $a_c$ is substitute by $a_e$ and geometry function was in this case [20]:

$$Y(\alpha_c) = \frac{1.99 - \alpha_c(1 - \alpha_c)(2.15 - 3.93\alpha_c + 2.70\alpha_c{}^2)}{(1 + 2\alpha_c)(1 - \alpha_c)^{3/2}}, \tag{13}$$

where $\alpha_c = a_c/W$.

The cohesive softening function has to be known to calculate cohesive fracture toughness $K_{Ic}{}^c$, which can be interpreted as an increase in the resistance to crack propagation caused by the bridging of aggregate grains and other toughening mechanisms in the FPZ. This function describes the relationship between the cohesive stress and effective crack opening displacement. Based on the author's previous studies, it can be stated that the type of softening function (bilinear or non-linear) had no significant effect on the calculated values of fracture parameters. The input parameters of the softening function are more important, especially the way of estimation of tensile strength. Commonly, the compressive strength values are used for the estimation of the tensile strength of the materials. It is more appropriate to determine the tensile strength directly from the tensile test performed in the configuration similar to the loading of the real structural element. Therefore, the tensile strength $f_{t,ID}$ was identified from the measured *F-d* diagrams (see Section 2.4.2). In this study, the results obtained using the non-linear softening function according to Hordijk [34] are presented. Then, the cohesive stress $\sigma(CTOD_c)$ at the tip of an initial notch at the critical state can be obtained from this softening function:

$$\sigma(CTOD_c) = f_t \left\{ \left[ 1 + \left( c_1 \frac{CTOD_c}{COD_c} \right)^3 \right] \exp\left( -c_2 \frac{CTOD_c}{COD_c} \right) - \frac{CTOD_c}{COD_c} \left( 1 + c_1{}^3 \right) \exp(-c_2) \right\} \tag{14}$$

where $f_t$ is the tensile strength, $CTOD_c$ is the critical crack tip opening displacement according to Jenq and Shah [35]:

$$CTOD_c = CMOD_c \left( \left( 1 - \frac{a_0}{a_c} \right)^2 + \left( 1.081 - 1.149 \frac{a_c}{W} \right) \left( \frac{a_0}{a_c} - \left( \frac{a_0}{a_c} \right)^2 \right) \right)^{\frac{1}{2}} \tag{15}$$

$COD_c$ is the critical crack opening displacement, and $c_1$ and $c_2$ are the material constants, which were taken from [36]. In this paper, $COD_c$ is calculated using a value of fracture energy $G_f$ determined using Equation (9) according to this formula [22]:

$$COD_c = \frac{5.136 G_f}{f_{t,ID}}. \tag{16}$$

Subsequently, the linear function for the calculation of cohesive stress $\sigma(x)$ along the length of the effective crack can be formulated [22]:

$$\sigma(x) = \sigma(CTOD_c) + \frac{x - a_0}{a_c - a_0} (f_t - \sigma(CTOD_c)). \tag{17}$$

When this relation is known, the cohesive fracture toughness $K_{Ic}{}^c$ is determined as follows [37]:

$$K_{Ic}^c = \int_{a_0/a_c}^{1} 2\sqrt{\frac{a_c}{\pi}} \sigma(U) F(U, \alpha) dU \tag{18}$$

where the substitution $U = x/a_c$ is used and $F(U,\alpha)$ is determined according to [37]

$$F(U, \alpha) = \frac{3.52(1 - U)}{(1 - \alpha)^{3/2}} - \frac{4.35 - 5.28U}{(1 - \alpha)^{1/2}} + \left( \frac{1.30 - 0.30U^{3/2}}{(1 - U^2)^{1/2}} + 0.83 - 1.76U \right) [1 - (1 - U)\alpha] \tag{19}$$

where $\alpha = a_c/W$.

The following formula based on the formerly obtained parameters was used to calculate the initial cracking toughness $K_{Ic}{}^{ini}$:

$$K_{Ic}^{ini} = K_{Ic}^{un} - K_{Ic}^c \tag{20}$$

where $K_{Ic}{}^{ini}$ represents the phase of stable crack propagation.

At last, the load level $F_{\text{ini}}$ expressing the load at the outset of stable crack propagation from the initial notch was determined according to this relation:

$$F_{\text{ini}} = \frac{4 \cdot S_M \cdot K_{\text{Ic}}^{\text{ini}}}{S \cdot Y(\alpha_0) \cdot \sqrt{a_0}} \tag{21}$$

where $S_M$ is the section modulus (calculated as $S_M = 1/6 \cdot B \cdot W^2$), $S$ is the span length, and $Y(\alpha_0)$ is the geometry function (Equation (13)); where $\alpha_0 = a_0/W$ is used instead of $\alpha_c$.

### 2.5. Cyclic/Fatigue Fracture Tests

The 3PB tests were also carried out to determine the basic fatigue characteristics of the investigated composite with an alkali-activated slag matrix. The central edge notch with a nominal depth of about 1/10 of the specimen's height was made by a diamond blade saw. The span length was set to 120 mm. The specimens were tested at different ages of their hardening, namely between the age of 140 and 160 days. The cyclic fracture tests were conducted using the servo-hydraulic testing machine Zwick/Roell Amsler HC25 with the load range of 0−25 kN (Figure 6a). An arrangement of a 3PB cyclic fracture test configuration is shown in Figure 6b. The loading process was governed by a force; the force amplitude was controlled. The stress ratio $R = F_{F\min}/F_{F\max} = 0.1$, where $F_{F\min}$ and $F_{F\max}$ refer to the minimum and maximum load of a sinusoidal wave in each cycle. The load frequency was set to 10 Hz. The number of cycles before failure was recorded for each specimen. The specimens were loaded in the range of high-cycle fatigue from $10^3$ to $10^6$ number of cycles. Therefore, the upper limit to the number of cycles $N$ to be applied was selected as 2 million cycles. The test was finished when the failure of the specimen occurred or the upper limit of loading cycles was reached, whichever occurred first.

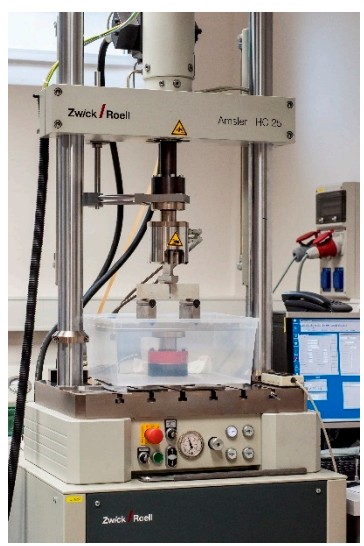

(**a**)

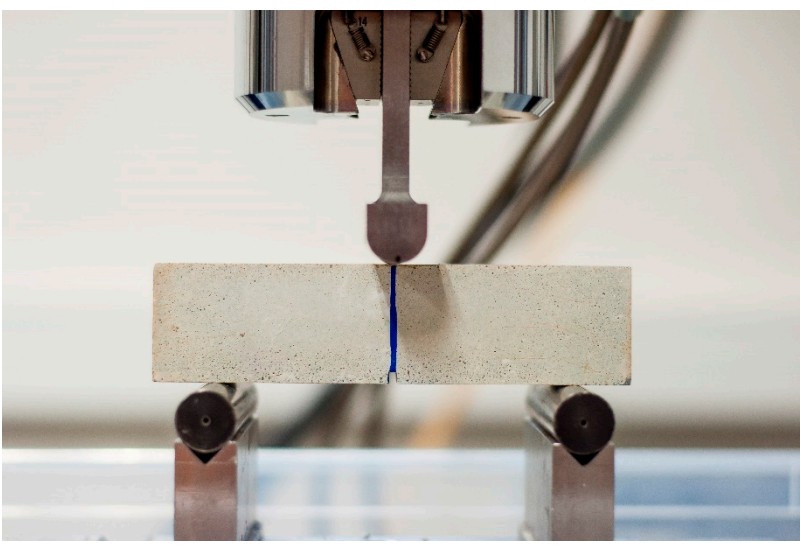

(**b**)

**Figure 6.** The testing machine used for cyclic/fatigue fracture test (**a**); the arrangement of test (**b**).

The results of the cyclic fracture test are presented in the form of an empirically derived $S_F−N$ curve, which is known as the Wöhler curve [4]:

$$S_F = a \cdot N^b \tag{22}$$

where $S_F$ is the maximum stress in 3 PB, $N$ is the number of cycles, and $a$ and $b$ are material constants. The maximum stress $S_F$ is calculated as follows:

$$S_F = \frac{3F_{F\max}S}{2B(W - a_0)^2}.$$ (23)

The parameters of the $S_F$–$N$ curve are determined only for test specimens that were broken during the cyclic test, and the specimens that withstand 2 million cycles are not taken into consideration. The fatigue limit was determined as the highest stress level at which three test specimens withstand 2 million cycles.

### 3. Results and Discussion

The results of performed experiments that involved determining physical, mechanical, and mechanical fracture parameters are displayed in Figures 7 and 8. The mean values accompanied by the sample standard deviations (error bars) are shown for all investigated characteristics. Although the test specimens were produced in three different batches, the variability of results is low, which proves the possibility to produce the composite with reproducible properties. As a result of the non-traditional curing conditions of ambient temperature and high RH $\geq$ 95% for a long period, the results of strength, fracture, and fatigue characteristics are hard to compare with other AAS or ordinary Portland cement (OPC) materials. For this reason, the results are partially compared in terms of materials of similar composition cured under different conditions.

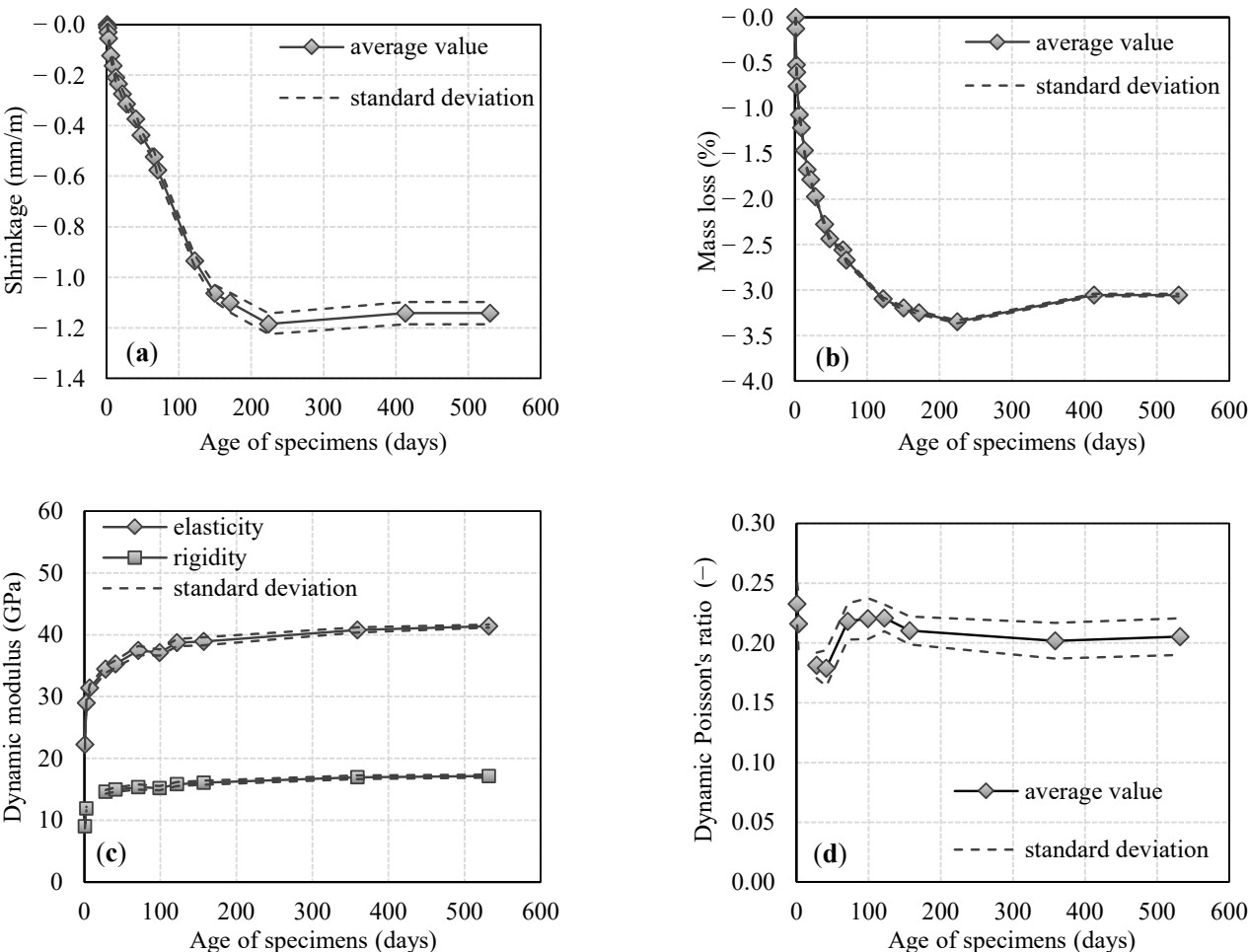

**Figure 7.** The mean values of shrinkage (**a**), mass loss (**b**) at RH of (55 $\pm$ 10)%, dynamic modulus of elasticity and rigidity (**c**), and dynamic Poisson's ratio (**d**) at RH $\geq$ 95% of the investigated alkali-activated slag (AAS) composite during the specimen's aging.

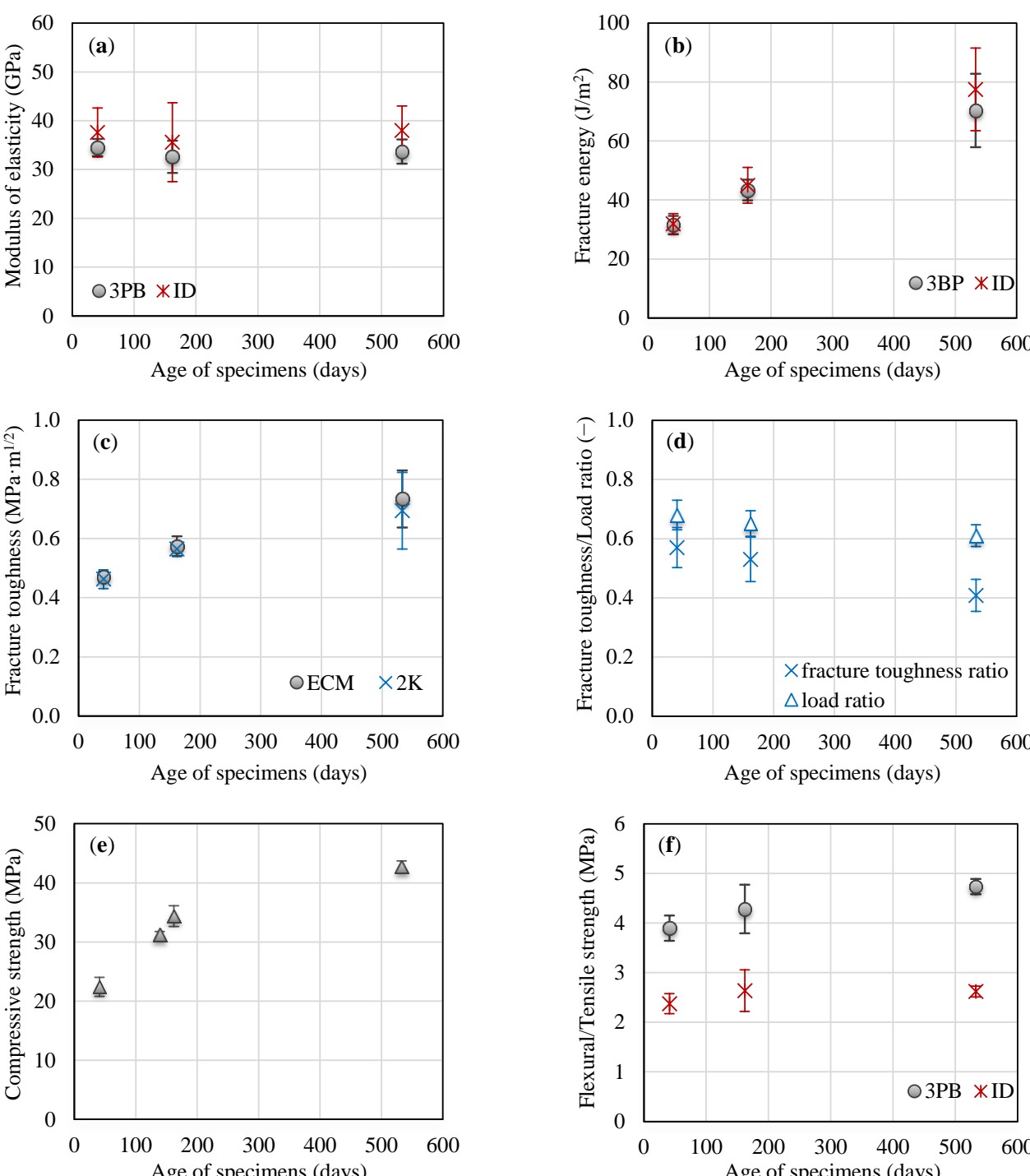

**Figure 8.** The mean values of modulus of elasticity (**a**), fracture energy (**b**), fracture toughness (**c**), fracture toughness/load ratio (**d**), compressive strength (**e**), and flexural/tensile strength (**f**) of the investigated AAS composite during the specimen's aging (3PB—direct evaluation of particular parameter from fracture test, ID—identification of particular parameter, ECM—effective crack model, 2K—double-*K* fracture model).

### 3.1. Basic Physical and Mechanical Parameters

The shrinkage measurement was performed in regular intervals on the three same specimens during the whole period of the specimen's aging. The results are presented in Figure 7a. The steady-state value of shrinkage of 1.2 mm/m was recorded at the age of about 200 days. The trend of the shrinkage process is similar to the shrinkage process in

OPC materials [38,39]. The difference is observed in time when the shrinkage stabilizes, which is two times later than in the OPC of similar composition cured under the same conditions. The absolute value of shrinkage of the investigated AAS composite at the age of 90 days is about two times higher in comparison with an OPC material with a similar water to binder (w/b) ratio and the same sand to binder (s/b) ratio cured under the same conditions [39]. On the other hand, the shrinkage is about six times lower in comparison with an AAS composite activated by waterglass with the similar w/b ratio, the same s/b ratio, and a higher activator dose stored under the same curing conditions [40].

The values of the dynamic modulus of elasticity are determined using the resonance method. At least three specimens are used to determine this parameter for a particular age of specimens. The value of the dynamic modulus of elasticity is about 40 GPa at the specimen's age of 530 days (see Figure 7c). The significant increase of this parameter is observed during the first 40 days of hardening when it achieves 35 GPa, but the follow-up increase with the specimen's age is not so high. A similar trend of development of the parameter value during hardening is also observed in the case of dynamic modulus of rigidity, which is about 17 GPa at the specimen's age of 530 days (see Figure 7c).

The dynamic modulus of elasticity and rigidity of the investigated AAS composite at the age of 530 days are about 1.5 and 1.8 times higher, respectively, in comparison with an AAS composite activated by waterglass with a similar w/b ratio and the same s/b ratio [41]. Nevertheless, the AAS composite activated by waterglass was stored at a lower RH of about 60%, i.e., under more severe conditions with the risk of microcracking and limited hydration process. The dynamic modulus of elasticity and rigidity of the investigated AAS composite at the age of 28 days is about 20% lower in comparison with OPC materials with a w/b ratio of 0.40 and the same s/b ratio, cured under the same conditions.

The Poisson's ratio was around 0.21 during the whole time of aging (see Figure 7d). A higher fluctuation was observed during the first 70 days when the dynamic modulus of elasticity and rigidity exhibited a steep increase. All dynamic characteristics gradually stabilized after this age. The absolute value of Poisson's ratio is similar to OPC materials of similar composition cured under the same conditions.

### 3.2. Mechanical Fracture Parameters

The mechanical fracture parameters determined during the specimen's hardening based on direct evaluation of records of fracture test in the form of the $F-d$ and $F-CMOD$ diagrams are presented in this section (Figure 8). These results are supplemented by parameters determined based on identification using an ensemble of ANNs. The number of specimens taken into the account for parameters evaluation is evident from Figure 5 where only the correctly recorded diagrams are presented. The results obtained by direct evaluation and identification are marked as 3PB and ID, respectively.

The value of modulus of elasticity determined by direct evaluation is about 33.5 GPa and is the same during the whole period of hardening (see Figure 8a) when the variability of the results is taken into account. The modulus of elasticity determined by identification is about 3 GPa higher for all investigated ages of specimens. The variability of results is higher for values determined by the method of identification. The modulus of elasticity of the investigated AAS composite at the age of 90 days is slightly higher in comparison with the OPC material with a similar w/b ratio and the same sand s/b ratio [39]. The modulus of elasticity at the age of 530 days is more than two times higher in comparison with the AAS composite activated by waterglass [42]. Nevertheless, the AAS composite activated by waterglass was stored at a lower RH of about 60%, i.e., in more severe conditions with the risk of microcracking and limited hydration process.

The fracture energy value gradually increases during the whole period of hardening (see Figure 8b). The highest value of about 70 J/m$^2$ is observed at the specimen's age of 530 days, which reaches more than 200% and 60% higher values in comparison with value at the age of 40 and 160 days, respectively. The same trend of development of this parameter during the hardening is observed for both the direct evaluation and identification. In the

case of the AAS composite, the stability loss during the loading did not occur; therefore, the post-peak parts of $F{-}d$ diagrams are taken into consideration when the fracture energy is calculated, unlike the OPC material with a similar w/b ratio and the same s/b ratio [39].

The fracture toughness value gradually increases during the whole period of hardening (see Figure 8c). The same values were obtained for both used fracture models. The highest value of about 0.73 MPa·m$^{1/2}$ is observed at the specimen's age of 530 days, which reaches about 56% and 28% higher values in comparison with value at the age of 40 and 160 days, respectively. A similar value of fracture toughness is observed for the AAS composite activated by a combination of sodium carbonate and hydroxide [42]; however, only the value at the age of 28 days is presented in this literature. The fracture toughness of the investigated AAS composite at the age of 90 days is about 30% lower in comparison with the OPC material with a similar w/b ratio and the same s/b ratio [39]. Nevertheless, both composites were stored at a lower RH of about 55−60%.

The resistance to stable crack propagation is represented by fracture toughness ratio: the initial cracking toughness $K_{Ic}^{ini}$ to unstable fracture toughness $K_{Ic}^{un}$ ratio. The fracture toughness ratio gradually decreases during the period of composite hardening (see Figure 8d). A similar trend is also observed for load ratio, which represents the ratio of load $F_{ini}$ at the outset of stable crack propagation to the maximum load $F_{max}$. A little bit lower fracture toughness ratio value is observed for the AAS composite activated by a combination of sodium carbonate and hydroxide stored at a lower RH of about 60% [43].

The compressive strength value gradually increases during the whole period of hardening (see Figure 8e). The value at the age of 530 days about 43 MPa is almost twofold in comparison with the value at the age of 40 days. The compressive strength of the investigated AAS composite at the age of 90 days is comparable with the OPC materials with a similar w/b ratio and the same s/b ratio stored at lower RH [39] and slightly lower than for the AAS composite with the similar w/b and s/b ratio and the same activator dose stored at a lower RH of about 65% [24]. It is necessary to emphasize that the 28 days' compressive strength of AAS with sodium hydroxide is about 70% lower than in the case of OPC materials of similar composition cured under RH $\geq$ 95%. While OPC materials show a significant increase in strength during the first 28 days, which makes it possible to use the 28 days' compressive strength for reliable structural design, in the case of AAS materials, the increase in strength is much slower and takes a longer time. It is evident from Figure 8e that in the case of the investigated AAS composite, the steep increase in compressive strength is observed up to the age of about a half-year. It is necessary to respect this fact during structural design where the long-term characteristics should be considered for effective and reliable design.

The flexural strength also gradually increases during the whole period of hardening (see Figure 8f); however, the increase is not so significant as in the case of compressive strength. The value at the age of 530 days about 4.7 MPa is about 20% higher in comparison with the value at the age of 40 days. A similar trend is also observed for tensile strength obtained by identification, but the values are equal to about 60% of the flexural strength. The flexural strength of the investigated AAS composite at the age of 90 days is slightly lower in comparison with the AAS composite with a similar w/b and s/b ratio and same activator dose stored at a lower RH of about 65% [24]. The same ratio of flexural to compressive strength of about 0.13 was observed [24].

### 3.3. Fatigue Characteristics

The results of cyclic fracture tests of specimens in 3PB at different load levels are summarized in Figure 9a in the form of a Wöhler curve (Equation (22)) where the maximum normal stress $S_F$ (in Figure 9 marked as *y*) applied during experiments is plotted against the logarithm of the number of cycles to failure $N$ (in Figure 9 marked as *x*). Theoretically, all test specimens are broken after the same number of cycles for one particular stress level. However, the fatigue behavior of a heterogeneous material such as the investigated AAS composite is distant from an optimal case, and the results show variability. It is also

connected with the different ages of specimens during the cyclic tests. In this case, the specimens were tested between the age of 140 and 160 days. It is evident that mechanical fracture parameters of the investigated AAS composite significantly change during the specimen's aging (see Figure 8), even after 160 days of hardening.

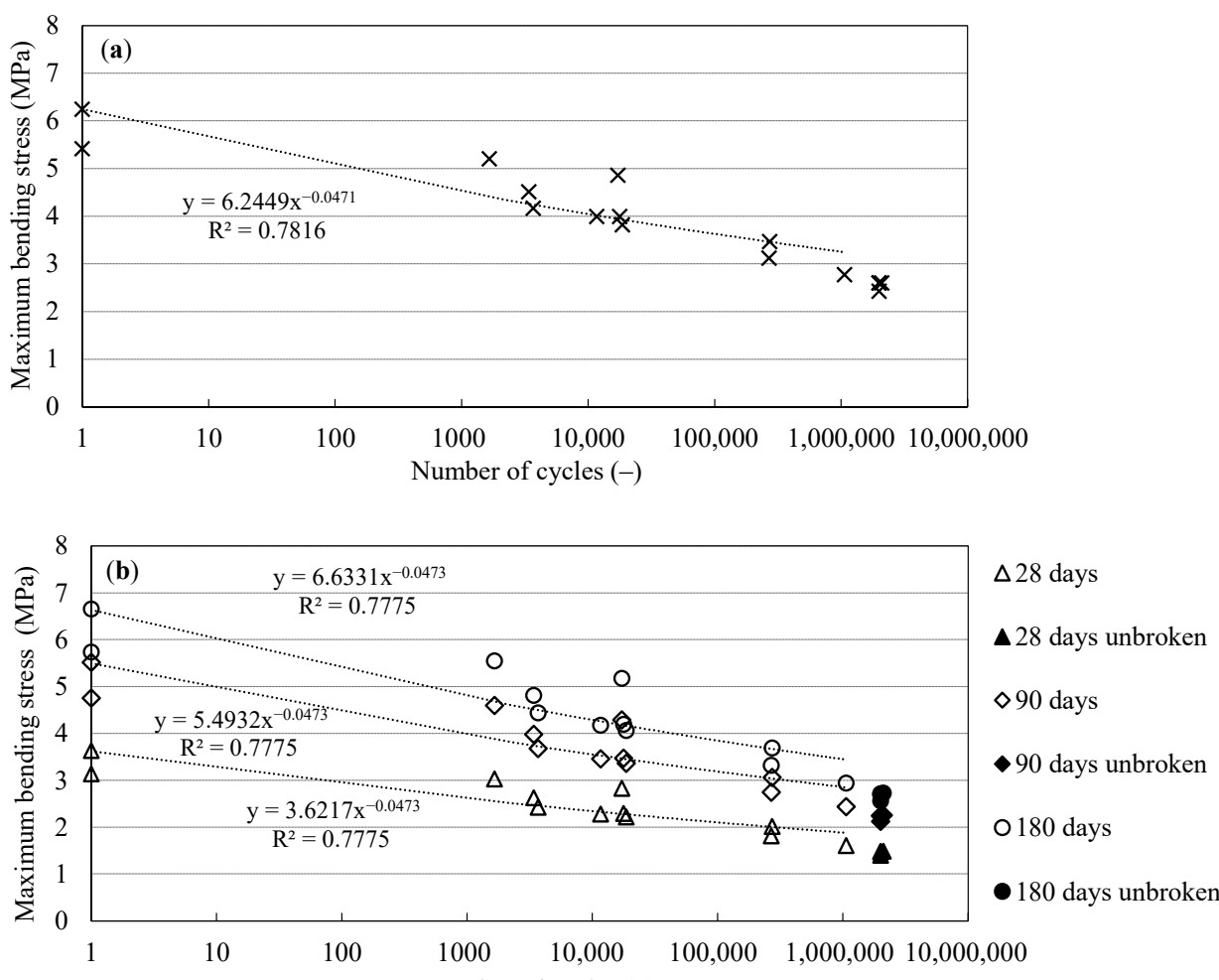

**Figure 9.** The $S_F-N$ curve for the investigated AAS composite plotted for measured data (**a**), the $S_F-N$ curve plotted for standardized data at the specimen's age 28, 90, and 180 days (**b**); the horizontal axis is plotted in logarithmic scale.

Therefore, the measured data are divided by coefficients determined from the approximation curve of relative compressive strength values to obtain correct values of fatigue characteristics corresponding to the age of specimens at which a particular cyclic test is performed. The measured data are standardized to a particular age of specimens by this procedure. In this case, the compressive strength is chosen because it is the most commonly determined mechanical parameter of quasi-brittle composites, and it is used as an input parameter in the structural design. The compressive strength test is also less time-consuming in comparison with the static fracture test. Nevertheless, the presented procedure can be used with any other mechanical or fracture parameter.

The measured values of compressive strength are divided by the average value at the chosen specimen's age. In this case, the compressive strength values at the specimen's age of 28, 90, and 180 days are chosen. The compressive strength at the age of 28 days is usually used as a reference value in the design of cement-based composites. The other two ages of specimens are chosen because the mechanical fracture parameters of the investigated AAS composite significantly change during the specimen's aging. By this procedure, the relative

compressive strength values for all investigated ages of specimens are obtained. Then, these values are approximated by the selected function:

$$y = a \cdot \left(1 - e^{-b \cdot x^c}\right),$$ (24)

where $x$ is the specimen's age in days, $y$ is the dimensionless relative compressive strength, $a$ is the coefficient corresponding with an asymptote to the approximation curve, in other words, the ratio of the theoretical value of the compressive strength at infinitum to the value of compressive strength at the specimen's age of 28, 90, or 180 days, and $b$, $c$ are the coefficients corresponding with the size of the time-dependent change of compressive strength, which is generally dependent on the compositions of the used mixture. The approximation is performed using the GTDiPS software [28]. The procedure is based on the non-linear least-square method provided by genetic algorithms, which are implemented in this open-source Java GA package [28]. The approximation curves and coefficients $a$, $b$, and $c$ for relative compressive strength values related to the specimen's age of 28, 90, and 180 days are displayed in Figure 10.

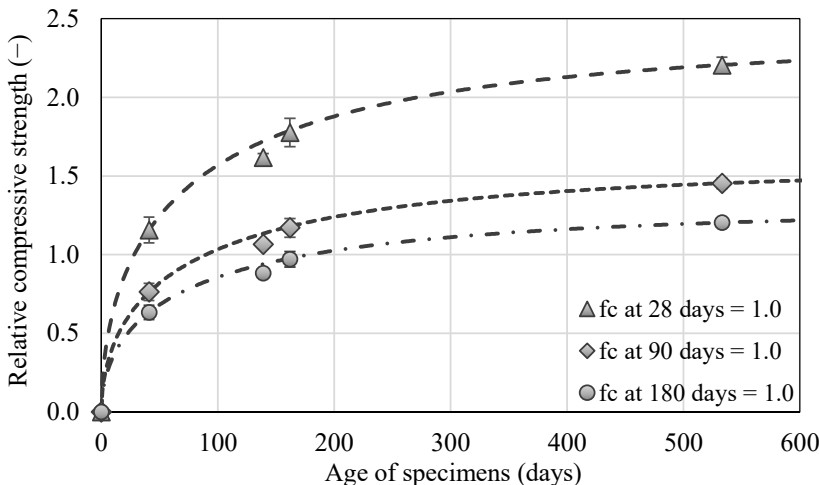

**Figure 10.** The approximation curves for relative compressive strength values related to the specimen's age of 28, 90, and 180 days, respectively.

The Wöhler curves for standardized data are introduced in Figure 9b. In addition, the fatigue limit was determined as the highest stress level at which three test specimens withstand 2 million cycles, and it is equal to 1.48, 2.24, and 2.71 MPa for data standardized using relative compressive strength values related to the specimen's age of 28, 90, and 180 days (see Figure 9b). The fatigue parameters of quasi-brittle materials are not commonly determined, although many structures made of these materials are often subjected not only to static but also to repetitive cyclic loads. If only the measured data without the effect of the particular age of specimens during the cyclic test are compared, then it might seem from Figure 9a that the investigated AAS composite has slightly better fatigue behavior than the OPC material with the same s/b and w/b ratios stored in the water [44]. In fact, the OPC material has slightly better fatigue behavior when the age of specimens is taken into the account and standardized data to the age of 28 days are compared with Figure 9b. The fatigue limit is about 0.5 MPa higher in the case of the OPC material [44].

## 4. Conclusions

The main aim of the present research is to investigate the long-term mechanical, fracture, and fatigue behavior of a fine-grained composite based on sodium hydroxide-activated slag under RH $\geq$ 95%. The structures made of quasi-brittle materials are usually loaded in a mixed-mode manner in civil engineering applications; therefore, the attention in the performed experimental campaign was focused also on this issue. The results presented

in this article were used as valuable input parameters of material models used for numerical simulations of crack propagation in the semi-circular specimens loaded under mixed-mode I/II conditions. The results of numerical simulations were used for the verification of experimentally obtained data and were presented in a separate article [17].

The obtained results can be used in the design of quasi-brittle materials with the alternative binder to commonly used Portland cement. The presented results are valuable primarily because the parameters are monitored from a long-term point of view, and the complex set of material characteristics was collected as well. The attention was also focused on the fracture and fatigue parameters that are not commonly investigated. These parameters give more comprehensive information about material behavior and could be utilized in the effective design of newly developing materials. Although the test specimens were produced in three different batches, the variability of results is low, which proves the possibility to produce the composite with reproducible properties.

In addition, a substantial long-term increase in mechanical fracture parameters such as fracture energy, fracture toughness, or compressive strength proves that AAS hydration considerably proceeds even within the years. Therefore, the identification of these parameters only at relatively early stages, such as after 28 days, can lead to underestimation of their full potential. It simultaneously amplifies the role of appropriate numerical simulations as an alternative to the long-term testing.

**Author Contributions:** Conceptualization, H.Š., B.K., V.B. and L.M.; methodology, H.Š., B.K. and V.B.; software, M.L.; validation, P.M., and L.M.; formal analysis, H.Š. and B.K.; investigation, H.Š. and B.K.; resources, H.Š., B.K. and V.B.; data curation, H.Š., B.K. and M.L.; writing—original draft preparation, H.Š., B.K. and V.B.; writing—review and editing, L.M., P.M. and M.L.; visualization, H.Š.; supervision, H.Š. and L.M.; project administration, H.Š.; funding acquisition, H.Š. and L.M. All authors have read and agreed to the published version of the manuscript.

**Funding:** This research was funded by the Czech Science Foundation, grant number 18-12289Y.

**Institutional Review Board Statement:** Not applicable.

**Informed Consent Statement:** Not applicable.

**Data Availability Statement:** Data are available on request from the corresponding author.

**Conflicts of Interest:** The authors declare no conflict of interest.

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
