# Peer review of "Mechanical Fracture and Fatigue Characteristics of Fine-Grained Composite Based on Sodium Hydroxide-Activated Slag Cured under High Relative Humidity"

_applsci, doi:10.3390/app11010259_

Round 1

Reviewer 1 Report

This paper is a useful characterization of mechanical properties from an alkali-activated composite cured with elevated humidity (RH=95%). However, before acceptance requires minor changes, whose suggestions are described below:

  1. The value of relative humidity during curing is a determining value in the mechanical properties since it prevents the appearance of cracks that reduce these properties. The title should include the value of relative humidity during curing in tests.
  2. Mechanical properties were tested with a relative humidity of 95%, then this material is not valid for in situ structures, only for precast elements. Introduction should include this fact.
  3. 4.1. Equation (15). Values of c1 and c2 were taken from [36], but this was lightweight and normalweight concrete. The difference between relative humidity during curing should be justified.
  4. 4.1. Equation (16), (18), and (19). The origin of these equations is not clear.
  5. Results and discussion. In this research, specimens have been made of a single alkaline cement mixture that is compared with the results of other publications with other compositions. The graphs only include the results obtained from this mixture, so it is poor. All graphs (figure 7, 8, 9, and 10) should include the values of the samples to which they are compared from the literature indicating its composition and curing conditions.
  6. 11, line 359. Missing reference to OPC specimen shrinkage values.
  7. Figure 7. Shrinkage was tested with RH=55% and dynamic modulus of elasticity, dynamic Poisson´s ratio was tested with RH=95%. This fact should be indicated in the caption.
  8. 14, line 448. The high relative humidity to which the samples have cured makes it necessary to indicate that the structural design is only for precast elements.
  9. Pg. 17, line 515. Relative humidity during curing should be indicated.
  10. 17, line 521. Paper [17] is not published, is under revision.

Author Response

We would like to thank the reviewers for their time spent on our manuscript, and for their constructive comments. We have responded to them below, as well as by modifying the revised article (using the "Track Changes").

Ad 1) The information about the relative humidity was added to the article title (page 1, line 4).

Ad 2) The information about this fact was added to Introduction (page 4, line 137).

Ad 3) Because it was not possible to perform the uniaxial tensile test to obtain the relationship between the cohesive stress and effective crack opening displacement, therefore the constants for quasi-brittle material with similar tensile strength were taken from literature.

Ad 4) These equations were taken from the literature, the respective references were added to the article. The equation (16) follows from the nonlinear softening relation given by Eq. (14).

Ad 5) The article summarizes a wide range of physical and mechanical characteristics of a specific type of composite material. We believe that the display of only the results of the performed experiments is beneficial for the clarity of the paper. The results obtained for the investigated AAS composite are compared with results already published in conference papers or journal articles and are properly cited. This way of comparison of results complies with common publication practice.

Ad 6) The reference was added (page 11, line 361).

  1. Kucharczyková, B., Topolář, L., Daněk, P., Kocáb, D., Misák, P. Comprehensive Testing Techniques for the Measurement of Shrinkage and Structural Changes of Fine-Grained Cement-Based Composites during Ageing. Adv Mater Sci Eng 2017, Vol. 2017, Article No. 3832072. https://doi.org/10.1155/2017/3832072

Ad 7) The information was added in the caption of Figure 7 (page 12, line 390).

Ad 8) This comment was addressed by adding of following statement in the Introduction:

In practical applications of investigated material, these non-traditional curing conditions can be found in the design of precast elements rather than in in-situ applications.

Ad 9) The information was added to the article (page 18, line 522).

Ad 10) The revised manuscript of the mentioned article was already sent to publisher. The statement was changed in the Introduction (page 3, line 95) and also in the Conlusions (page 18, line 528):

The detailed information can be found in a separate article, which is currently under consideration for publication [17].

Reviewer 2 Report

The authors illustrated the results of a comprehensive experimental campaign regarding the long-term mechanical behavior and fracture of the sodium hydroxide activated slag fine-grained composite. The topic is of current interest in the scientific debate on how to use new concretes in civil engineering. The experimental research is exhaustive, especially regarding the analysis of the fracture and its numerical simulation. Furthermore, the study of the variation of the mechanical characteristics under cyclic fatigue stress is in-depth. This method of analysis is also applicable to other diverse types of cement mixtures. However, before acceptance requires minor changes, whose suggestions are described below:

  • It would be desirable by the authors to investigate the variation in mechanical performance after freeze-thaw cycles, since even this environmental action, which is also very frequent in the useful life of any civil structures, over time can imply strong variations in the mechanical performance of the material, especially regarding the elastic modulus. Have the authors taken into consideration the execution of freeze-thaw tests, or at least evaluated from the scientific literature the implications that these actions may have on the durability of the tested material?
  • Is it possible to evaluate the fatigue of the material with faster tests, i.e. with a lower number of cycles even without breaking to standardize the experimental procedure?
  • Explain in more detail in what type of structures this innovative cement mixture can be used and what the advantages in terms of durability can be compared to traditional cement mixtures.

Author Response

We would like to thank the reviewers for their time spent on our manuscript, and for their constructive comments. We have responded to them below, as well as by modifying the revised article (using the "Track Changes").

Ad 1) Freeze-thaw tests of the same fine-grained composite (and its comparison with other alkali-activated as well as a cement-based mixture) is a part of our other research focused on the durability of these materials. In general, alkali-activated materials achieve excellent durability, but the specific issue of the presented fine-grained composite based on sodium hydroxide-activated slag is its gradual long-term evolution of mechanical parameters, as is illustrated in the manuscript. If this mortar is tested too early before reaching sufficient mechanical properties, e.g. after 28 days, its freeze-thaw resistance is poor. Nevertheless, this manuscript is not focused on freeze-thaw resistance, but long-term characteristics in a stable, well-defined environment. The results can serve as a reference to any other conditions as well as contribute to designing of appropriate testing methodology for alkali-activated materials, which is a very current, widely discussed issue.

Ad 2) The fatigue experiments were performed in the range of high-cycle fatigue as defined by [4] because it corresponds to real cyclic loading which can occur during the designed structure’s service lifetime such as automotive and train traffic, machine vibration, and wind action. The range for high-cycle fatigue is 10^3 to 10^6. Therefore, it is appropriate to perform also the tests for a higher number of cycles.

The information about the range of high-cycle fatigue was added (Page 10, line 335)

Ad 3) Unlike our other project mentioned within the first response, this manuscript is not focused on durability. Compared to traditional cement mixtures, alkali-activated based are advantageous especially in resistance to aggressive environments, which could potentially be taken into account for their application in practice (structures exposed to salts, acids, or in chemical productions). The advantage of the tested mixture lies also in terms of low shrinkage and cracking potential at lower relative humidity, compared to those where other alkaline activators are used. In practical applications of investigated material, the non-traditional curing conditions (high relative humidity) used in our experiments can be found in the design of precast elements rather than in in-situ applications. This information was added in the Introduction.